# Aligned Nanofiber-Guided Bone Regeneration Barrier Incorporated with Equine Bone-Derived Hydroxyapatite for Alveolar Bone Regeneration

**DOI:** 10.3390/polym13010060

**Published:** 2020-12-25

**Authors:** Jae Woon Lim, Kyoung Je Jang, Hyunmok Son, Sangbae Park, Jae Eun Kim, Hong Bae Kim, Hoon Seonwoo, Yun Hoon Choung, Myung Chul Lee, Jong Hoon Chung

**Affiliations:** 1Department of Biosystems Engineering, Seoul National University, Seoul 08826, Korea; jwlim1130@snu.ac.kr (J.W.L.); shmking@snu.ac.kr (H.S.); sb92park@snu.ac.kr (S.P.); Je6740@snu.ac.kr (J.E.K.); ser21@hanmail.net (H.B.K.); 2Division of Agro-System Engineering, Gyeongsang National University, Jinju 52828, Korea; trudwp@gmail.com; 3Department of Industrial Machinery Engineering, College of Life Science and Natural Resources, Sunchon National University, Sunchon 57922, Korea; uhun906@scnu.ac.kr; 4Department of Otolaryngology, Ajou University School of Medicine, Suwon 16499, Korea; yhc@ajou.ac.kr; 5Bk21 Plus Research Center for Biomedical Sciences, Ajou University Graduate School of Medicine, Suwon 16499, Korea; 6Research Institute of Agriculture and Life Sciences, Seoul National University, Seoul 08826, Korea

**Keywords:** osteogenesis, equine bone, hydroxyapatite, nanofiber, regeneration barrier

## Abstract

Post-surgery failure of dental implants due to alveolar bone loss is currently critical, disturbing the quality of life of senior dental patients. To overcome this problem, bioceramic or bone graft material is loaded into the defect. However, connective tissue invasion instead of osteogenic tissue limits bone tissue regeneration. The guided bone regeneration concept was adapted to solve this problem and still has room for improvements, such as biochemical similarity or oriented structure. In this article, an aligned electrospun-guided bone regeneration barrier with xenograft equine bone-derived nano hydroxyapatite (EBNH-RB) was fabricated by electrospinning EBNH/PCL solution on high-speed rotating drum collector and fiber characterization, viability and differentiation enhancing properties of mesenchymal dental pulp stem cell on the barrier was determined. EBNH-RB showed biochemical and structural similarity to natural bone tissue electron microscopy image analysis and x-ray diffractometer analysis, and had a significantly better effect in promoting osteogenesis based on the increased bioceramic content by promoting cell viability, calcium deposition and osteogenic marker expression, suggesting that they can be successfully applied to regenerate alveolar bone as a guided bone regeneration barrier.

## 1. Introduction

The human organs and tissues are developed in aligned matter. Tendon [1] and muscle [2] are such examples, and even bone tissue [3] exists in an aligned manner as well. Aligned structures in tissues serve many purposes, providing physical strength and unique functions. Moreover, it is well known that providing micro- to nanoscale aligned topographic structures and physical strength, as in vivo, can enhance the activity of the cells grown above. The aligned structure makes cells elongate in a certain direction, which leads to genes or signaling pathways activation of cell proliferation or adhesion [4]. Previous studies reported that the alignment of surface topography can also have synergetic effects with the physicochemical properties of surface materials to promote osteogenic differentiation of cells, which can be considered an appropriate strategy to adapt in many tissue engineering examples [5,6,7,8].

The importance of alveolar bone regeneration is rising because it is one of the critical elements in dental implant treatment. Dental implants can serve an excellent masticating function and have many advantages leading to patient satisfaction when compared to conventional methods such as dentures. Nevertheless, post-surgery failures are constantly reported. If a tooth is lost, the surrounding bones decrease in volume and density. This results in vulnerability of the implant site, which is not a rare case [9,10].

Thus, bone regeneration is essential in dental implant surgery. The bone graft material used for this purpose is bioceramic and can promote bone regeneration, and hydroxyapatite-based synthetic or xenograft product is commonly used [11,12,13].

Usually in powder form and having an artificial or natural origin, hydroxyapatite-based bone graft material has biocompatibility, biochemical similarity, osteoconductive and osteointegrating nature [14,15,16,17,18,19,20]. Xenograft materials or products such as Bio-Oss^®^ showed osteointegration effects from in vivo tests. However, bone graft material or bioceramics themselves are brittle, have low impact resistance and relatively low tensile strengths, thus are hard to apply at the load bearing sites without additional solution [21]. Additionally, the invasion of connective tissues and cells into defects during the healing process is a major problem. When bioceramics are applied without any further treatment, little or no sign of bone healing or ingrowth of connective tissues are observed in the defect [22] or loss of original graft volume [23], resulting in delayed bone fusion or immature ossified condition [23,24].

The guided bone regeneration (GBR) concept can be a solution. GBR is the method to isolate bone graft implant sites and gum tissue by placing a physical, biocompatible barrier between the sites [25,26]. The purpose of this strategy is to prevent invasion of nearby connective tissues and allow bone tissue-dominated regeneration. In many cases, it has been reported that the barrier is effective using only a bone graft material. Nevertheless, the solvent cast film-type polymer barrier still has some room for improvement since it is porous but does not have much surface area, brittle and inflexible without cytotoxic plasticizer [27] leading to less binding site [28], little or minimum growth and proliferation enhancement [29] to bone progenitor cells growing above. A fabricating method that can provide a polymer membrane with a high surface area and binding site and enhance cell growth would be suitable for the purpose.

Electrospinning presents some solutions to overcome the disadvantages of GBR films. It is a popular, relatively simple method to produce nano- to micro-diameter fiber mats that have a high surface area and controllable porosity due to a networked structure that also provides many binding sites to cells, including osteogenic sites. A simple change in the collector can also provide an aligned, oriented surface of nanofibers, such as in vivo tissue structures; the biodegradability depends on polymer variation [30,31,32,33]. By simply including bioceramics with sub/nanosized particles in a polymer solution, biochemically osteo-similar fibers can be produced that can target synergetic effects that might contribute to the osteogenic differentiation signaling pathway of mesenchymal stem cells [34,35]. FDA-approved poly(ε-caprolactone) can be the right polymer for fabricating GBR barriers, having many attractive features, such as biocompatibility, biodegradability, and low pricing, compared to other well-used polymers, such as poly(lactic acid) (PLA) or poly(lactic-co-glycolic acid) (PLGA) [32,33].

In advance of previous reports that included hydroxyapatite (HA)-based bone graft material in electrospun mats, xenograft equine bone-derived nanohydroxyapatite (EBNH) was selected for this study. EBNH is produced by sintering equine bone at a high temperature, which is free of an immune response, infection, and bovine prion-associated disease when compared to conventional bovine xenograft bone graft materials [36] and has advantages such as biocompatibility and osteogenic activity promotion [37].

In this study, we produced aligned, EBNH-incorporated electrospun PCL-guided bone regeneration barriers (EBNH-RBs), which are biocompatible and biochemically similar to natural bone tissue. Finally, dental pulp-originated mesenchymal stem cells (DPSCs), which have multipotency and are capable of differentiating into dental tissues, including alveolar bone [38,39], were cultured on the scaffold in vitro to determine its biocompatibility and osteogenic differentiation-promoting abilities on osteogenic cells. (Figure 1).

The objectives of this study are (1) to fabricate novel, aligned bone generation barrier with EBNH and (2) to enhance osteogenic differentiation of dental pulp stem cells by synergistic effects of fiber alignment and EBNH.

## 2. Materials and Methods

### 2.1. Materials

PCL (Mw 80,000), nHA, and Alizarin red staining agent (Aldrich, Gillingham, UK) were purchased from Sigma-Aldrich. WST-1 solution was purchased from Goma Biotech. Chloroform and dimethylformamide (Daejung, Gyeonggi-do, Korea) were purchased from Duksan. Culture media, antibiotics, fetal bovine serum (FBS), and Dulbecco’s phosphate-buffered saline (DPBS) (Welgene, Daegu, Korea) were purchased from Welgene. Dental pulp stem cells were kindly provided by the School of Dentistry, Seoul National University. Polyclonal anti-human osteopontin goat IgG, anti-goat IgG fluorescein isothiocyanate (FITC)-conjugated, tetramethylrhodamine (TRITC)-conjugated phalloidin, and 4′,6-diamidino-2-phenylindole (DAPI) (R&D systems, Minneapolis, MN, USA) were purchased from R&D Systems. Alexa Fluor^®^ 594 (Thermo Fisher, Frederick, MD, USA) was purchased from Thermo Fisher.

Equine bone-derived nanohydroxyapatite (EBNH) was prepared from previous methods as follows: washed, H_2_O_2_-pretreated horse bone was sintered at 1300 °C in a furnace to remove organic components and possible contaminants, and then the remaining minerals were processed with a blade grinder and high-power ball mill to produce finer particles and sieved for even size distribution [33].

### 2.2. Electrospinning Process

A 10% PCL (*w*/*v*) solution was prepared by chloroform and DMF as the solvent. For each group, EBNH and nHA were equivalently mixed into PCL solution following 10 s of 3 sets of sonication. The composition of each group is listed in Table 1.

A continuous flow syringe pump (KD scientific KDS270, Holliston, MA, USA) was used to feed PCL solution at a constant rate. Nanofibers were collected on a metal collector at a 100 mm nozzle to ground metal collector distance, 23 °C, 30 to 40% moisture, 22-gauge nozzle, 18 kV voltage, 0 mA current, and syringe pump rate of 0.6 mL/h for 30 min. All solutions were sonicated for 3 sets of 10 s to disperse the solute evenly. The aligned group was collected under the same conditions, except for a voltage of 12 kV and grounded rotating drum collector spinning at 3000 RPM considering the fiber collection rate and alignment. The remaining solvent was removed by ventilation under a fume hood overnight. The resulting fiber mat was punched on the PDMS layer to make an in vitro sample and stored for further use.

### 2.3. Fiber Characterization

#### 2.3.1. Scanning Electronic Microscopy (SEM)

The sample was placed in a 60 °C dry oven for at least one day to remove the remaining solvent and moisture, sputter coated, and placed on a grid, and images were taken using a field emission scanning electron microscope (Zeiss SUPRA 55 VP, Oberkochen, Germany).

#### 2.3.2. Fiber Orientation Analysis

The fiber orientation of the samples was examined by the OrientationJ plugin of ImageJ 1.51 k. Eight random sites were selected from the SEM image of each sample.

#### 2.3.3. X-ray Diffractometry (XRD)

Four variations of the sample (Control, 1% nHA, 1% EBNH, 10% EBNH) were tested using an X-ray diffraction instrument (Bruker D8 Advance, Karlsruhe, Germany) in settings of Cu-K 3 kW.

### 2.4. In Vitro Experiments

Based on a literature study, four in vitro groups were selected: PCL, nHA, 1% EBNH, and 10% EBNH, which all have aligned fiber topography. Before seeding, EBNH-RB were sterilized with 70% ethanol and then washed with DPBS three times. Then, the scaffolds were exposed to UV light for 30 min to remove any possible contaminants. Sterilized scaffolds were put into well plates, and the cells were pre-seeded for at least 1 h to prevent unwanted growth on the TCPS. Then, α-MEM with 10% fetal bovine serum and 0.5% antibiotics was supplied and changed on a regular schedule. Osteogenic media was supplied after cell attachment in the differentiation assay, which was composed of 0.1  μM dexamethasone, 10 μM
β–glycerophosphate, and 100  μM ascorbic acid added to the proliferation media.

#### 2.4.1. Cell Viability Assay

To examine the viability of EBNH-RBs, $1\times 10^{3}$ cells/well were seeded onto 6 mm samples in 96-well culture plates (*n* = 5). For the procedure, WST-1 agent was diluted in proliferation media as described in the manufacturer’s instructions. Each well was aspirated and washed three times with DPBS. Two hundred microliters of WST solution was placed into each well, and then the plate was cultured at 37 °C for an hour. Then, 100 µL of solution from each well was transferred to a new 96-well plate and measured in an ELISA machine at 450 nm wavelength. The resulting data were converted as percentages compared to the control (PCL Aligned).

#### 2.4.2. Alizarin Red Assay

To examine the mineral deposition of cells on EBNH-RBs due to osteogenic differentiation, $3\times 10^{3}$ cells/well were seeded onto 96-well culture plates (*n* = 7) supplied with osteogenic media. After a pre-determined period, each well was aspirated and washed three times with DPBS and then fixed with 4% paraformaldehyde solution. Then, each well was washed with DPBS three times. Then, 200 µL of the Alizarin red solution was put into each well, and the plate was cultured at room temperature for 1 h. Next, solution from each well was aspirated, and 200 µL of destaining solution was put into each well and cultured at room temperature for 1 h to harvest the dyes absorbed by the mineralized tissues. One hundred microliters of the destained fluid was transferred to a new 96-well plate and measured by an ELISA at 570 nm wavelength.

#### 2.4.3. Immunocytochemistry

To determine the expression of osteogenic markers, $3\times 10^{4}$ passage 5 dental pulp cells were seeded onto 24 mm diameter nanofiber-PDMS scaffolds in 24-well plates (*n* = 5), and other procedures, such as scaffold sterilization and fixation, were performed as described above.

After fixation and washing three times with 0.01% Tween 20-PBS solution (PBST), 1% Triton X solution was added to each well for 15 min to permeabilize. After washing three times with PBST, 1% bovine serum albumin (BSA) solution was added to each well and incubated at room temperature for 1 h for blocking. Appropriate concentrations of primary antibody (polyclonal anti-human osteopontin goat IgG, R&D Systems AF1433), secondary antibody (anti-goat IgG fluorescein isothiocyanate (FITC)-conjugated, tetramethylrhodamine (TRITC)-conjugated phalloidin), Alexa Fluor^®^ 594 and DAPI solution were continuously added for 1 h, except 5 min for DAPI. Each sample was washed three times with PBST between and after treatment. All scaffolds were mounted on a cover glass after staining was performed and imaged with a high-resolution fluorescence microscope (Nicon N-Storm, Tokyo, Japan).

The intensity of each image was analyzed with the ROI manager tool in ImageJ, randomly selecting five 50,000 pixel-sized square image sites.

### 2.5. Statistical Analysis

The statistical significance of quantitative data was analyzed by performing Duncan’s multiple range test and two-way ANOVA between each group. At *p* < 0.05, differences between groups were considered to be statistically significant.

## 3. Results and Discussion

### 3.1. Characterization of EBNH-RBs

As shown in Figure 2, the addition of EBNH and nHA changed the individual fibers on a rougher surface but nevertheless maintained sub-micro- to nanodiameters and unique microstructures (Figure 2a). In fiber orientation analysis by OrientationJ, aligned scaffolds showed a distribution of orientations concentrated over 1000 in certain degree orientations, while random patterned scaffolds showed a distribution of orientations under 1000 with varying degree orientations (Figure 2b).

Almost every EBNH-RB showed HA with a similar XRD peak at 2θ = 22.5 and 25 except for the PCL-only group. These results show that EBNH maintained HA-like chemical properties even after being electrospun with PCL (Figure 2c). These features may have interacted with the alignment of fiber to induce synergetic activity for the mesenchymal stem cells, which give oriented microstructure and biochemical properties to EBNH-RBs making it similar to in vivo osteogenic tissue. It is well known that cells can change morphology by the cultured surface, and nanofibers are one of the examples. Cell lines grown on aligned fibers tended to have spindle-like cell morphology even at the single-cell level. [39]. Adding biominerals such as HA causes the aligned fiber structure with an encapsulated or coated structure to resemble the microstructure of the bone or dentin [40]. There is a report that aligned nanofibers with HA can induce osteogenic differentiation even without any soluble factors [41], which may explain the cell behavior of DPSCs cultured on EBNH-RBs.

### 3.2. Cell Viability

In the WST results, groups with a higher ratio of bioceramics showed a larger percentage of cell viability in vitro. The 10% EBNH group had a significantly larger percentage of viability than the nHA and 1% EBNH groups on day 3 and day 7. Especially on day 7, the viability of the 10% EBNH group increased almost two-fold (Figure 2d).

Although PCL itself is a biocompatible material, a rise in the EBNH ratio resulted in a significant increase in viability. This can be evidence that the alignment of PCL fibers and bioceramics may have some synergistic effect, enhancing cell adhesion, viability, and other related features. An increase in EBNH concentration could have induced the microstructure to have more in vivo-like properties along with alignment, therefore increasing the affinity and viability of the EBNH-RBs to DPSCs. The addition of HA can also decrease crystallinity by disrupting the growth of the crystal surface, and decreased crystallinity leads to increased cell viability [42]. EBNH may disrupt crystal surface growth more than nHA, especially at high concentrations, which leads to higher DPSC viability. Additionally, the fiber surface in the rough state could be an outcome of surface disruption.

### 3.3. Osteogenic Differentiation

Groups with aligned fiber structures and bioceramics showed stronger figment colors than the controls, and 10% EBNH showed a significantly larger value than the control in week 4 (Figure 3a).

Immunocytochemistry results show osteogenic marker protein osteopontin (OPN) present in the scaffold after differentiation, presenting aligned patterns and the actin cytoskeleton in bright red while nuclei of DPSCs are present in a blue color stain. (Figure 3b).

Osteogenic marker protein expression showed the osteogenic differentiation enhancing effect of EBNH more clearly. The 1% and 10% EBNH group showed significantly higher OPN expression than the nHA group, and 10% EBNH group showed significantly higher OPN expression even when compared to control. This is quite controversial from the week 4 Alizarin red result. Thus, at least for OPN expression, EBNH is better than nHA and can enhance overall features, including cell viability, mineral deposition, and osteogenic marker expression, in a higher ratio than nHA.

Previous studies have noted that aligned structures can enhance cell adhesion to the surface, and aligned structures can have a synergistic effect with the physicochemical properties of the material. PCL and EBNH, which are the two main components in this study, showed excellent incorporation into each other and maintained unique chemical properties. While the microstructure and biochemical composition of EBNH-RBs is similar to the in vivo minimal unit of bone or osteon, the distribution of the graft material in the fibers may have decreased surface crystallinity and increased cell-surface affinity, leading to higher OPN expression, as explained above in the viability data. EBNH-RBs are expected to enhance the osteogenic differentiation of mesenchymal cells by synergistic effects between EBNH and the aligned structure, which is expected to serve as an excellent barrier for alveolar bone-guided regeneration, while the application to bone defects in another defect makes the platform versatile.

## 4. Conclusions

A novel, aligned bone generation barrier with EBNH (EBNH-RB) was fabricated by electrospinning, and the barrier enhanced osteogenic differentiation of mesenchymal stem cells by synergistic effects of fiber alignment and EBNH. The results suggest that alignment of PCL and EBNH has a synergistic effect in enhancing the growth, viability, and overall osteogenic differentiation of mesenchymal stem cells, especially in the PCL with 10% EBNH group (Figure 4). EBNH-RB would be excellent platform for promoting bone regeneration effect on dental implant site as guided bone regeneration barrier, and growth factors or other bone regeneration promoting elements can be adapted to the barrier as well for further in vivo experiment and actual therapeutic application.

## Figures and Tables

**Figure 1 polymers-13-00060-f001:**
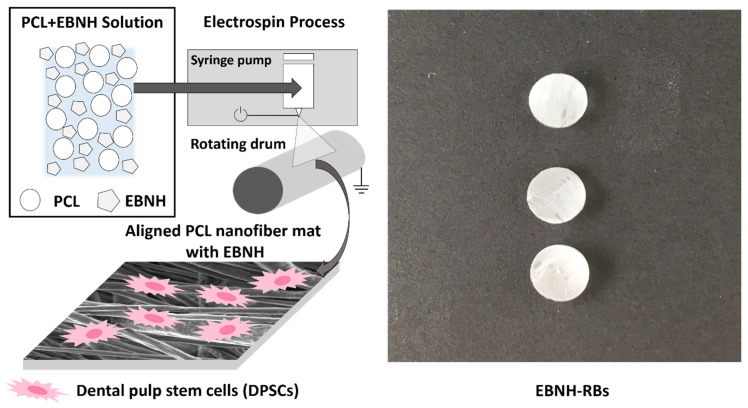
Overall scheme of the research. Abbreviations: PCL, poly(ε-caprolactone), EBNH, equine bone-derived nanohydroxyapatite.

**Figure 2 polymers-13-00060-f002:**
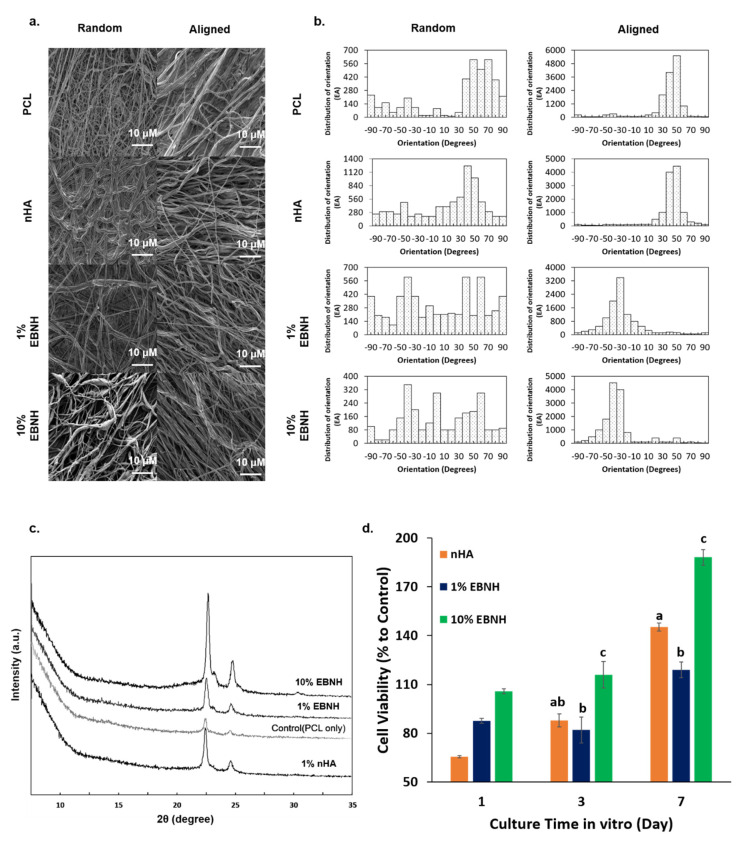
Characteristics of EBNH‑RBs. (**a**) FESEM images of the fiber. (**b**) Distribution of fiber alignment. Aligned EBNH‑RBs showed at least 3000 to 4000 distributions in certain alignments compared to random EBNH‑RBs. (**c**) XRD peaks of electrospun nanofibers. (**d**) WST‑1 results of cells cultured on EBNH‑RBs by day. EBNH-RBs maintained the peaks of EBNH or hydroxyapatite after electrospinning, and viability increased by almost two-fold in the day 7 10% EBNH group. (Having non-identical letters, such as a and b, means that the groups have statistically significant differences at *p* < 0.05 (*n* = 5).

**Figure 3 polymers-13-00060-f003:**
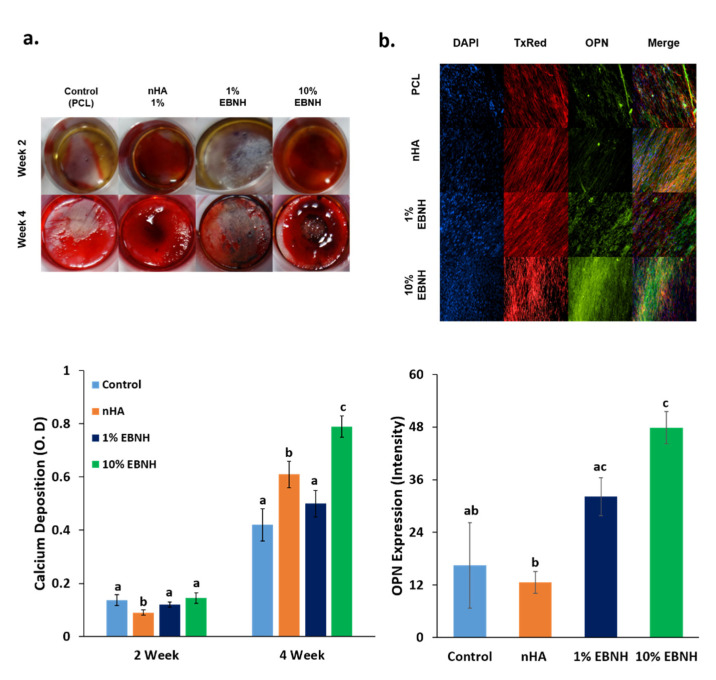
Differentiation of cells on EBNH-RBs. (**a**) Alizarin red staining images and destain results of dental pulp stem cells on nanofiber scaffolds by week. (**b**) Immunocytochemistry (ICC) fluorescent image of dental pulp stem cells 4 weeks after differentiation. DAPI stained the cell nucleus, TxRed (Alexa 594) stained the actin cytoskeleton, and FITC-TRITC conjugated phalloidin stained OPN. (Having non-identical letters, such as a and b, means that the groups have statistically significant differences at *p* < 0.05 (*n* = 5). Groups with common letters to others, such as a and ab, were not significantly different. While the 10% EBNH groups had a significantly high or high tendency in the differentiation assays, it is remarkable that the 1% EBNH group had less calcium deposition but expressed higher levels of OPN than the nHA group.

**Figure 4 polymers-13-00060-f004:**
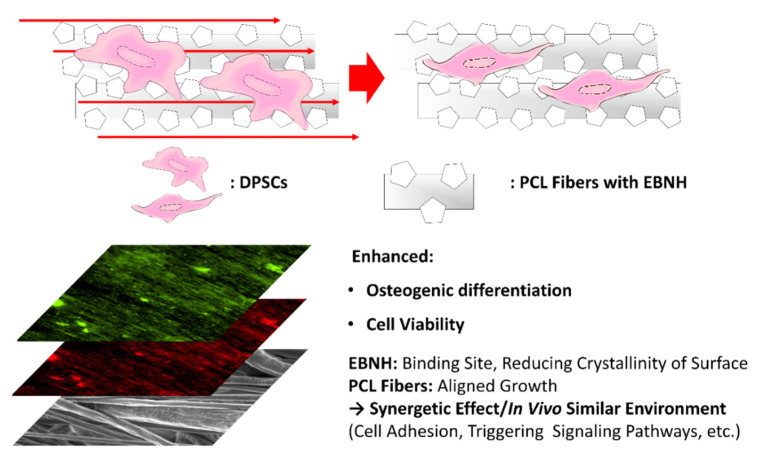
Overall summary of the study. EBNH-RBs enhanced osteogenic differentiation and cell viability, possibly by the synergetic effect of EBNH, reducing crystallinity and alignment of PCL nanofibers combined with EBNH to induce a similar in vivo environment.

**Table 1 polymers-13-00060-t001:** Composition of experimental groups and control.

Name	Bioceramic (mg)	PCL (g)
**Control (PCL)**	0 mg	1 g
**nHA**	10 mg (nHA)	1 g
**1% EBNH**	10 mg (EBNH)	1 g
**10% EBNH**	100 mg (EBNH)	1 g

## Data Availability

All the experimental data herein presented are made available upon request to the corresponding author.

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
