# Peer review of "Aligned Nanofiber-Guided Bone Regeneration Barrier Incorporated with Equine Bone-Derived Hydroxyapatite for Alveolar Bone Regeneration"

_polymers, 2020, doi:10.3390/polym13010060_

Round 1
Reviewer 1 Report
ALIGNED NANOFIBER-GUIDED BONE REGENERATION BARRIER INCORPORATED WITH EQUINE BONE-DERIVED HYDROXYAPATITE FOR ALVEOLAR BONE REGENERATION
Manuscript ID: polymers-1031095
Polymers
The aim of the present in vitro study was to investigate the osteogenic potency of xenograft equine-bone-derived nanohydroxyapatite incorporated electrospun PCL produced barrier.
Even though the topic has already been largely discussed in the scientific literature and the study design is not original, the obtained results are very interesting and clearly illustrated.
Moreover, the manuscript in its current form has weaknesses and should be improved as suggested below.
Introduction
- To include a reference about this sentence: "However, bone graft material or bioceramics themselves have poor physical properties since they are fine particles, often in the slurry or putty phase when in use." Please cite references at the end of each sentence not just at the end of each paragraph to make it clear which information is derived from a particular source.
- The following sentences should be provided in a clearer fashion: " When bioceramics are applied without any further treatment, tissues other than bone are observed in the regenerated tissue, resulting in functional and structural defects." and "Nevertheless, the film-type barrier still has some room for improvement since it cannot provide an oriented structure or physical stimulation and misses many biocompatible elements, such as porosity, binding sites, and biodegradability, in some materials". Moreover, Authors should better discuss, also quoting scientific literature, the same sentence.
- The following sentence is a redundancy: "The morphology and characteristics of the tissue were determined by various methods."
Material & Methods
- Add trademark, city and state for all used materials.
Results and Discussion
- 3.3 Osteogenic differentiation section: "In the intensity analysis of OPN 231 fluorescence, 10% EBNH-RB did not show significantly higher OPN expression than 1% EBNH-RB, 232 while an increase in OPN expression did exist in the other groups." should be amended as " In the intensity analysis of OPN 231 fluorescence, 10% EBNH-RB did not show significantly higher OPN expression than 1% EBNH-RB, 232 while an increase in OPN expression did not exist in the other groups."
Figures
- In Fig. 2 a the magnification should be added.
- In Fig. 3: the a) caption should be corrected as " Alizarin red staining images and destain results of dental pulp stem cells on nanofiber scaffolds by week."; In which image is it possible to see " .....viability increased by almost two-fold in the day 7 10% EBNH group"?; " While the 10% EBNH groups had a significantly high or high tendency in the differentiation assays, it is remarkable that the 1% EBNH group had less calcium deposition but expressed higher levels of OPN than the nHA group." Significantly high or high tendency? What do you mean by tendency?
- In Fig. 4 " Overall summary of the study. EBNH-RBs enhanced osteogenic differentiation and cell viability, possibly by the synergetic effect of EBNH, reducing crystallinity and alignment of PCL nanofibers combined with EBNH to induce a similar in vivo environment." should be edited as " Overall summary of the study. EBNH-RBs enhanced osteogenic differentiation and cell viability, possibly by the synergetic effect of EBNH, reducing crystallinity and alignment of PCL nanofibers combined with EBNH to induce a similar in vivo environment."
Reference
- More references should be added about the osteoconductive and osteoinductive properties of bioceramic and xenograft materials.
Once the suggested minor revision has been undertaken, publication of the present manuscript is strongly recommended.
Reviewer 2 Report
The manuscript titled “Aligned Nanofiber-guided Bone Regeneration Barrier Incorporated with Equine Bone-derived Hydroxyapatite for Alveolar Bone Regeneration” submitted to “Polymers” for publication: investigated an aligned electrospun-guided bone regeneration barrier with xenograft equine bone-derived nano hydroxyapatite (EBNH-RB) osteogenesis promotion targeting the dental implant applications. This is a well-designed and well conducted study and the manuscript fits well within the scope of the journal; it needs some major improvements; there are a few suggestions that authors may consider to improve it further:
The use of English language is reasonable, however, there are a number of punctuation and grammatical errors; that should be corrected and rephrased using academic English for a better flow of text for reader.
Abstract: should be improved; firstly, it is too brief. Secondly only covers background and aim of the study, it can be improved by adding some key methods and results.
Line 27: the abbreviation should not be repeated.
Introduction:
Lines 34-35: I am not convinced by the statements; could authors please modify or support these statements by authentic reference citations.
Line 50: although authors mentioned about the bone regeneration; the aspect of osseointergration and bioactivity should be addressed in context of dental implants; following most relevant articles can be included for this information
Bioactive surface coatings for enhancing osseointegration of dental implants." Biomedical, therapeutic and clinical applications of bioactive glasses. Woodhead Publishing, 2019. 313-329.
Bioactivity and osseointegration of PEEK are inferior to those of titanium: a systematic review. Journal of Oral Implantology. 2016 Dec 1;42(6):512-6.
Customized therapeutic surface coatings for dental implants." Coatings 10.6 (2020): 568.
Figure 2: how it was defined; either the nanofibers were aligned or random? Only surface?
Results and discussion are well presented; it is a good job by authors.
Figure 4 should be dragged in to the manuscript instead of after conclusions
What are the future prospects of this short communication? What the reader should expect in future should be described.
